# Connection between Social Relationships and Basic Motor Competencies in Early Childhood

**DOI:** 10.3390/children8010053

**Published:** 2021-01-17

**Authors:** Christian Herrmann, Kathrin Bretz, Jürgen Kühnis, Harald Seelig, Roger Keller, Ilaria Ferrari

**Affiliations:** 1Research Group Exercise and Sport, Zurich University of Teacher Education, 8090 Zurich, Switzerland; kathrin.bretz@phzh.ch (K.B.); roger.keller@phzh.ch (R.K.); ilaria.ferrari@phzh.ch (I.F.); 2Expert Group Physical Education, Schwyz University of Teacher Education, 6410 Goldau, Switzerland; juergen.kuehnis@phsz.ch; 3Department of Sport, Exercise and Health, University of Basel, 4001 Basel, Switzerland; harald.seelig@unibas.ch

**Keywords:** preschool, kindergarten, sport, health, integration, fundamental movement skills, physical education

## Abstract

In preschool, children build new contacts and social relationships with other people. They learn to cooperate with their peers and communicate in groups. In addition to social relationships, basic motor competencies (in German: Motorische Basiskompetenzen (MOBAK)) are also seen as a central developmental goal in early childhood and are necessary for participation in the culture of sports and movement. The aim of this paper is to describe the connection between social relationships and basic motor competencies in early childhood. In this present study, the motor competencies of *N* = 548 preschool children (51% girls, M = 68.0 months, SD = 6.8) were tested in the competence areas of self-movement and object movement. The children’s perceived social relationships were recorded from teacher and parent perspectives. The results clearly show a connection between social relationships and motor competencies in early childhood, with a stronger connection observed in boys. This finding is relevant both from a developmental and a health-oriented perspective, as it points to a link between physical and mental health, as well as technical and interdisciplinary competencies, in early childhood.

## 1. Introduction

Preschoolers are seen as being at an important stage for physical, mental, and social development. In engaging with their environment, they constantly discover new connections, acquire new gross and fine motor skills, and build social relationships [1,2]. The development of motor competencies and social relationships with peers in early childhood is a core developmental task of this stage of life, whose mastery has a positive influence on children’s mental and physical health [3,4].

In Switzerland, children from the age of four attend preschool, which is part of the mandatory schooling. There are cantonal and language-specific curricula that pursue the same goals in terms of physical and social development. The school career is divided into different stages. The first stage includes two years of preschool, as well as first and second grade. For all children, preschool is an expanded social environment with new tasks and challenges. Developmental goals are defined for subject-specific competencies, e.g., physical education (the pupils can balance on a narrow surface, e.g., walk along a long bench) [5]. In addition to subject-specific competencies, interdisciplinary competencies (e.g., interpersonal relationship skills and emotion regulation) should also be acquired in physical education [6]. Interdisciplinary competencies contribute to all health-related topics, such as well-being [7].

### 1.1. Social Interactions and Peer Relationships

Preschool plays an important role as a space for social interactions, as children between the ages of four and seven years spend a large part of the day there. The relationships children form in early childhood to their closest role models remain important during the preschool years and are supplemented by further role models. Some children go to a day nursery or play group before entering preschool and have already gained experience with other children in the absence of their parents. Other children experience themselves as independent and autonomous for the first time and learn to develop their own social strategies and to behave cooperatively and pro-socially toward their peers [8]. The social competencies they acquire are important for establishing and maintaining friendships in peer groups [9,10,11] and are related to executive functions such as cognitive flexibility [12]. Popular children participate more frequently in games, find resolutions to conflicts, and can agree on rules together [10]. For children who are rejected or neglected, in contrast, peer relationships can be a strain and can lead to problems resolving conflicts with peers, learning new skills, and coping with demands at school [13]. These children gain less social experience and are more rarely integrated into activities at and outside school.

### 1.2. Development of Motor Competencies

Preschoolers are at a stage of development in which they continually develop and extend their basic motor repertoire in contexts of social interactions (e.g., preschool, peer group, and sport groups) and differentiate it for use in different situations. Basic motor forms (e.g., throwing, jumping, running, and balancing) improve through regular practice. As children grow older, they learn and automate even more complex movement sequences [14] and can combine them with one another [15]. This forms the basis for the development in later childhood and adolescence of sport-specific motor skills, which constitute the basic techniques for individual sports (e.g., a power throw in handball) [16]. Within this connection, motor development is influenced by physical (e.g., body weight) and mental characteristics (e.g., self-assessed competency). Against this backdrop, childhood may be seen as a significant stage in the development of motor competencies [17].

Motor competencies that children need to participate in the culture of sports and movement are called basic motor competencies [18]. They guarantee a basic capacity to engage in sports and also form the basis for the development of higher competency levels required in specific, primarily extracurricular, fields of athletic activity [19]. For example, children can only participate in ball sports if they are sufficiently confident in controlling a ball.

Connections between physical activity and motor performance dispositions are increasingly becoming a topic of inquiry in the health sciences (for an overview, see [17]). Physical activity encourages motor development, whereas a low motor performance level can lead to lower physical activity [20]. A varied movement and sports behavior has a positive impact on physical health, fosters mental well-being, and contributes to social integration [21,22].

### 1.3. Peer Relationships and Motor Competencies

In preschool, friends are playmates who have similar interests and engage in similar activities. At this age, children mainly make friends with children of the same gender. Girls engage more frequently in cooperative forms of play, while boys engage more frequently in individual forms of play. With boys in particular, the circle of friends develops “competitive behavior” [23] and the resulting potential for conflict. Athletic ability plays a significant role in peer groups of boys [24].

Children with good motor performances are more popular and better-integrated into peer groups than those with poorer motor performances [25,26]. Low motor competencies can lead to negative interpersonal (problems with peers) and intrapersonal (low self-assessment) consequences at the psychosocial level, which, in turn, has an impact on mental health in the form of a downward spiral, thus emphasizing the relationship between motor competence and metal health [27,28].

The connection between social relationships and motor development receives particularly little attention in preschoolers. As the majority of existing studies were conducted at the primary- and secondary-school levels, in the present study, we will shed light on the connection between friendships and motor competencies in early childhood. In doing so, we will examine the friendships of preschoolers as assessed by their parents and teachers. In addition, we will investigate potential gender differences.

## 2. Materials and Methods

In February/March 2020, we measured the basic motor competencies of preschoolers in the Swiss cantons Ticino and Nidwalden. In Switzerland, preschool is integrated into the school system as a two-year entrance stage for primary school. The educational and developmental goals of preschool, which is organized into homeroom and class lessons, are formulated in the curriculum Lehrplan 21 [5].

Parallel to this, we conducted a questionnaire among the children’s parents or legal guardians and their teachers. In contrast to previous studies utilizing questionnaires among the children’s classmates as a means of determining peer status, we measured the children’s social integration and their interpersonal relationship skills via the assessments of their teachers and legal guardians.

Participation in the study was voluntary, and the legal guardians and teachers of the children concerned were informed about the study in advance. The legal guardians submitted written declarations of consent. The study was approved by the cantonal departments of education and the preschool principals.

Our study fully conforms to the Declaration of Helsinki. Ethical review and approval were partly waived for this study, as no medical parameters were collected in the study. As the study was conducted during regular physical education, the approval of the study was the responsibility of the cantonal school authorities. Therefore, the legal and school-relevant ethical requirements were approved by the Directorate of Education, Culture and Sport of the canton of Ticino (Repubblica e Cantone Ticino Dipartimento dell′educazione, della cultura e dello sport) and Cantonal School Authority of the canton of Nidwalden (Amt für Volksschulen und Sport) and the local school managements of the participating primary schools. The children and their parents were informed about the general purpose of the school project and the study, the voluntary nature of the participation, and the anonymous handling of the data. Furthermore, parents provided informed consent, and children assented to participate.

### 2.1. Sample

In total, we wrote to the parents or legal guardians of 956 preschoolers in the cantons of Ticino and Nidwalden, 701 of which (response rate: 73.3%) gave their written consent to participate in the study and sent back the parent questionnaire. In the present study, we admitted children in an age range of 55 to 80 months and were thus able to include *N* = 548 preschoolers (M = 68.0 months, SD = 6.8, 50.9% boys) from 52 classes (average class size, *n* = 10.5) and 16 preschools in the convenience sample. The subsample from Ticino contained 36 classes with *n* = 354 children (M = 66.5 months, SD = 6.5, 51.4% boys), and the subsample from Nidwalden contained 16 classes with *n* = 194 children (M = 70.7 months, SD = 6.4, 50.0% boys). In this total sample, we were able to collect data from *n* = 499 children (M = 68.1 months, SD = 6.9, 50.3% boys) on their basic motor competencies. In Switzerland, preschool classes are relatively small, with approximately 15 to 20 children per class. The low number of children per class is due to the exclusion of children without parental consent or children who were not present on the survey day. We received assessments from the teachers (M = 40.8 years, SD = 11.6, 95% female teachers) for *n* = 541 children (M = 68.0 months, SD = 6.8, 50.6% boys). We received assessments from the parents (M = 37.1 years, SD = 8.2, 81% mothers) for *n* = 532 children (M = 68.0 months, SD = 6.8, 50.6% boys).

#### 2.1.1. Test Instruments and Data Collection

##### Children: Basic Motor Competencies (In German: Motorische Basiskompetenzen (MOBAK))

To measure the basic motor competencies in the preschools, we used the MOBAK-KG test instrument, which enables curricular valid and age-specific measurements of the competencies in preschool physical education lessons [29]. The instrument measures the competency domains self-movement (4 items) and object movement (4 items) (Table 1; for details, see [29]) and contains the basic requirements emphasized explicitly in the field of sporting activity as the elementary learning goals of physical education (e.g., [5]). Each test item describes a standardized task with corresponding evaluation criteria. The children had two attempts to try to achieve the test item requirements (no trial run). The two single attempts were rated on a dichotomous scale (0 = failed, 1 = successful), and the individual results were added up to form the final item score (0 points = no successful attempts, 1 point = one successful attempt, 2 points = two successful attempts). The test items throwing and catching were an exception to this rule. In these cases the children had six attempts each, and the number of successful attempts was recorded. Afterwards, 0–2 successful attempts were scored as 0 points, 3–4 successful attempts as 1 point, and 5–6 successful attempts as 2 points. For each competency domain, it is possible to achieve a maximum sum value of eight points (for details, see [29]).

The validation study [30] succeeded in confirming the psychometric quality and the assignment of the test items into the two basic motor competencies of self-movement and object movement by means of confirmatory factor analyses (*N* = 403, Comparative Fit Index (CFI) = 0.98, Root Mean Square Error of Approximation (RMSEA) = 0.044 [30]).

The data were collected in classes during a regular 45-min lesson. The classes were split up into small groups of three to four children each and led through the eight test stations by trained testers. The testers gave a standardized explanation and one demonstration of each test item.

##### Teachers: Social Integration (Perception of Inclusion (PIQ))

The children’s social integration was measured via the assessment of their teachers. For this purpose, we used the corresponding subscale of the perception of inclusion (PIQ) questionnaire [31]. The teachers assessed the children individually via four items (Table 1) on a four-point scale. The teachers were sent a questionnaire for each child in advance, along with the information on the study. They were asked to complete it before the MOBAK testing and bring it with them on the day of the test. The internal consistency of the scale is satisfactory, with a Cronbach’s alpha of 0.76.

##### Parents: Interpersonal Relationship Skills (In German: Kompetenzen und Interessen Von Kindern (KOMPIK))

The children’s interpersonal relationship skills [32,33] were measured via four items (Table 1), which the parents responded to on a five-point scale in a questionnaire. Moreover, they were asked to provide the child’s date of birth and gender. The parent questionnaire was handed out to the parents and collected again by the teachers in an envelope, along with the declaration of consent. The internal consistency of the scale is satisfactory, with a Cronbach’s alpha of 0.75.

### 2.2. Data Analysis

The data editing and descriptive analyses were conducted with SPSS 25 (IBM Corp, Armonk, NY, USA) [34]. Influences of the multilevel structure (students from different classes) were tested with the help of interclass correlations (ICC). The multivariate analyses were calculated with Mplus 8.3 [35]. Three consecutive models were calculated.

Model 1: To test the factorial validity of the test instruments and calculate the latent correlations between the two MOBAK factors of self-movement and object movement, social integration, and interpersonal relationship skills, we calculated a four-factor confirmatory factor analysis.

Model 2: As the MOBAK factors in particular are closely associated with age [30], we integrated age in months into model 1 as a covariate.

Model 3: To test the differences between boys and girls, we calculated model 2 separately in a multigroup model for boys and girls. In doing so, we set the measurement model to be invariant between the genders (factorial invariance [36]) while allowing variations between the genders in the structural model.

In all models, we treated the MOBAK test items as ordinal-scaled data and the questionnaire items as interval-scaled data. Accordingly, we applied the means- and variance-adjusted weighted least squares (WLSMV) estimator, which allows for a robust estimation even of non-normally distributed data.

We considered the dependencies within the multilevel structure (0.03 ≤ ICC ≤ 0.23; Table 2) in all models by correcting the standard error with the “type = complex” function for nested datasets implemented in Mplus. The goodness of fit of the models was assessed with the help of the fit indices proposed in the literature [37]. Effect sizes were interpreted as small (r > 0.10, β > 0.05), medium (r > 0.30, β > 0.25), and large (r > 0.50, β > 0.45) [38,39].

The three data collection methods (MOBAK testing (*n* = 499 children), teacher surveys (*n* = 541 children), and parent surveys (*n* = 532 children)) led to missing values of an unsystematic nature (missing at random). For example, several children were sick when the MOBAK tests were conducted. On account of the age and gender distributions, however, it may be assumed that this did not lead to any biases in the overall sample (*N* = 548). Accordingly, we estimated the missing values via the full information maximum likelihood (FIML) algorithm. The FIML procedure is a conservative and well-established procedure in educational research. The FIML procedure prevents bias in the sample composition by preventing a reduction in the sample size [40].

## 3. Results

In a comparison between the age groups, older children showed much better performances in the basic motor competencies than younger children. Older children received better assessments on social integration and relationship skills. A consideration of the sum values (Table 2) shows that girls performed better on self-movement, while boys did better on object movement. Moreover, the teachers rated the girls as more socially integrated, while there were no significant gender differences with regards to the relationship skills assessed by the parents.

The calculated ICC values (Table 2) indicated that a significant portion of the total variances in object movement (13%) and in social integration (23%) as assessed by the teachers may be attributed to class membership, while class membership was less significant for self-movement and for interpersonal relationship skills as assessed by the parents.

### Latent Structural Equation Models

Model 1: The four-factor confirmatory factor analysis to test for factorial validity achieved a good model fit (χ^2^ =122.22; degrees of freedom (*df*) = 98; *p* = 0.049; CFI = 0.97; RMSEA = 0.021; *N* = 548). The individual factor values of the test items are listed in Table 1. This result confirmed the psychometric quality of the tests in this age group. The correlation between self-movement and object movement was r = 0.77 (*p* < 0.001) and, between social integration and interpersonal relationship skills, r = 0.21 (*p* < 0.001). The two motor competence areas were significantly correlated with social integration (self-movement: r = 0.20, *p* < 0.01 and object movement: r = 0.20, *p* < 0.01) and with interpersonal relationship skills (self-movement: r = 0.17, *p* < 0.001 and object movement: r = 0.20, *p* < 0.001).

Model 2: The four-factor confirmatory factor analyses with age as a covariate also achieved a good model fit (χ^2^ =133.25; *df* = 110; *p* = 0.065; CFI = 0.971; RMSEA = 0.020; *N* = 548). Age showed clear correlations with the two basic motor competencies of self-movement and object movement. It showed weak correlations with social integration and interpersonal relationship skills.

When age was taken into account as a covariate, the partial intercorrelations between the latent factors were weaker but remained without exception at a significant level (Figure 1).

Model 3: Taking model 2 as a starting point, we tested for differences between boys and girls in a multigroup comparison. In doing so, we calculated the correlations between the latent factors, as well as with age as a covariate, for both genders separately. This multigroup model achieved a good fit (χ^2^ = 255.14: girls: χ^2^ =124.77, boys: χ^2^ = 130.37, *df* = 244; *p* = 0.29; CFI = 0.987; RMSEA = 0.013; *N* = 548: girls = 279, boys = 269), demonstrating that the measurement models are invariant between the genders and, therefore, that no differential item functioning (DIF) is present (Table 3). Accordingly, gender comparisons are permissible at both the manifest and the latent levels.

Table 3 shows that the correlation of the covariate age with the latent factors hardly differs between the genders. As already became clear in model 2, older children showed a higher level of motor performances and were assessed by their teachers and parents as being better socially integrated (cf. Table 2 and Figure 1).

By contrast, we did find gender-specific differences in the intercorrelations of the latent factors. The correlations between the basic motor competencies and social integration (as assessed by the teachers) or interpersonal relationship skills (as assessed by the parents) were only significant in the boys. The correlations of self-movement and object movement with social integration turned out to be high in the boys, at *r* = 0.39 (*p* < 0.001) and *r* = 0.47 (*p* < 0.001), respectively, whereas these correlations were not significant in the girls.

## 4. Discussion

The objective of the present study was to investigate the connection between basic motor competencies and social relationships in early childhood. To do so, we collected data on children’s social relationships, both from the perspective of their parents and from the perspective of their preschool teachers, to obtain differentiated insights from various contexts.

The children already showed gender-specific differences in motor performance levels at the young ages of 4.5–6.5 years. Whereas the boys showed significantly better performances on movements involving balls (object movement), the girls achieved better results on whole-body movements (self-movement) [30]. Moreover, the girls were assessed by the teachers as being better socially integrated than their male peers. As in previous studies [25,26], there were also significant correlations between the children’s basic motor competencies and their social integration. Children with better motor competencies were assessed by their teachers and parents as being better socially integrated than children of the same age with poorer basic motor competencies. This correlation was more pronounced in the boys. This result can possibly be attributed to the relevance of athletic ability in peer relationships, especially among boys [24].

Since we did not directly interview the children regarding their social integration, biases by teachers or parents are possible. The girls were generally rated better in social relationships (see Table 2). These high values and the associated reduction in variance could explain the lower correlations. Whether the high ratings of the girls are partly due to the perceptions of the predominantly female teachers (95%) can unfortunately not be clarified at this point due to the lack of male teachers. However, the instruments for external assessment by parents (KOMPIK) and teachers (PIQ) are established and economical instruments. Correlations between the number of peer nominations and teachers’ assessments have been demonstrated in various studies [31].

Due to the cross-sectional study design, however, it was not yet possible to identify the direction of causality. Accordingly, future longitudinal studies should focus on the extent to which a potential need for support in early childhood influences children’s motor and mental developments. The current state of research provides indications that early experiences playing have an impact on children’s later ability to integrate themselves into a community [41] and that persons with high motor competence are more likely to be able to participate in the culture of sports and movement during their lifespan and are, consequently, also more physically active than persons with low motor competence [19].

This also raises the pedagogical–didactic question of how to design a careful promotion of physical activity in preschool. Due to the intertwined nature of development and learning in early childhood, curricula like the Swiss Lehrplan 21 formulate a combination of subject-specific and general competencies as developmental goals of the first cycle [5]. In this way, great importance is attached to social actions in groups, which opens up diverse social interactions and takes into account the heterogeneous conditions and needs of individual children.

As for the teachers, their task is to implement a varied and age-appropriate promotion of basic motor competencies. In addition to targeted forms of play in which the children can make intensive and highly variable use of basic motor competencies in various learning environments, they need to develop competency-oriented tasks (learning tasks) that aim at cooperative problem solving and in which even children with poor motor competence can play an active and productive role [42]. The intention here is to prevent the exclusion of these children and, at the same time, to create a situation in which those children experience the joy of movement and take the opportunity to improve their motor competence without feeling ashamed. In this context, extracurricular measures should also be examined and developed, such as the design of schoolyards that promote physical activity or the organization of extracurricular sports-oriented activities that can, among other things, provide a meaningful rhythm to everyday school life.

## Figures and Tables

**Figure 1 children-08-00053-f001:**
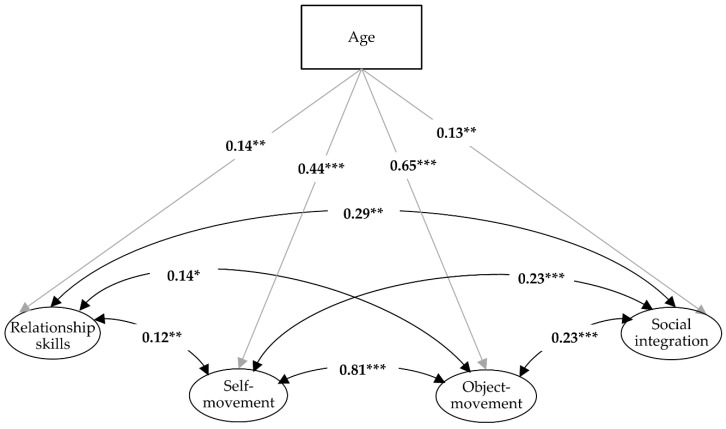
Four-factor confirmatory analyses with the covariate age (Model 2: * *p* ≤ 0.05, ** *p* ≤ 0.01, and *** *p* < 0.001).

**Table 1 children-08-00053-t001:** Description of the test items and their factor values (FV) in model 1. PIQ: perception of inclusion.

**Self-Movement ^a^** [29]		**FV**
Balancing	The child walks across an overturned long bench without stepping off the bench.	0.67
Rolling	The child performs a forward roll down an inclined mat and is able to land fluently in a standing position on his/her feet.	0.41
Jumping	The child continuously jumps a distance of 3.0 m on one foot, turns around, and jumps back 3.0 m on the other foot.	0.69
Running	The child runs forward along a corridor (0.6 m × 4.0 m) to a wall, touches it with his/her hand, and then runs back backwards.	0.67
**Object Movement ^a^** [29]		**FV**
Throwing	The child throws six juggling balls at a target of 1.1 m height at a distance of 1.5 m with overhead throws.	0.45
Catching	The tester drops a small basketball to the ground from a height of 1.5 m so that the ball bounces back up at least 1.1 m from the ground. The child catches the ball after it reaches the highest point.	0.75
Bouncing	The child bounces a small volleyball with both hands and catches it again without losing the ball.	0.71
Dribbling	The child dribbles a futsal ball through a marked corridor (2.8 m × 9.0 m) around two obstacles without stopping or losing the ball.	0.57
**Social Integration ^b^ (from the Teacher’s Perspective)** [31]		**FV**
PIQ 1	He/she has a lot of friends in his/her class.	0.81
PIQ 2	He/she gets along very well with his/her classmates.	0.70
PIQ 3	He/she feels alone in his/her class (-).	0.42
PIQ 4	He/she has very good relationships with his/her classmates.	0.48
**Interpersonal Relationship Skills ^c^ (from the Parent’s Perspective)** [32]		**FV**
KOMPIK 1	Your child plays with many different children (is not restricted to individual children).	0.65
KOMPIK 2	Your child is sought after as a playmate.	0.90
KOMPIK 3	Your child has close friendships with other children.	0.39
KOMPIK 4	Your child is important to other children, has influence in the group	0.69

Notes: KOMPIK: kompetenzen und interessen von kindern (in German); ^a^ 0 points, 1 point, 2 points; ^b^ 0 = strongly disagree, 1 = disagree, 2 = agree, and 3 = strongly agree, (-) = reversed coding; ^c^ 0 = disagree, 1 = somewhat disagree, 2 = somewhat agree, 3 = mostly agree, and 4 = agree.

**Table 2 children-08-00053-t002:** Descriptive sum values of the motor competency domains (0–8), social integration (0–12), and interpersonal relationship skills (0–16) by gender and age groups. ICC: interclass correlations.

Factors	Overall			Boys		Girls		55–67 Months		68–80 Months	
	M	95% CI	ICC	M	95% CI	M	95% CI	M	95% CI	M	95% CI
Self-movement	4.1	(3.9–4.3)	0.03	3.8	(3.5–4.1)	4.4	(4.1–4.7)	3.5	(3.2–3.8)	4.6	(4.3–4.8)
Object movement	4.2	(4.0–4.4)	0.13	4.7	(4.4–4.9)	3.8	(3.5–4.0)	3.1	(2.9–3.4)	5.1	(4.8–5.3)
Social integration	9.7	(9.6–9.9)	0.23	9.5	(9.3–9.7)	9.9	(9.7–10.2)	9.4	(9.2–9.7)	9.9	(9.7–10.2)
Relationship skills	11.0	(10.8–11.2)	0.06	10.9	(10.5–11.2)	11.1	(10.8–11.4)	10.5	(10.2–10.9)	11.3	(11.0–11.6)

Notes: M = Mean, 95% CI = 95% Confidence Interval.

**Table 3 children-08-00053-t003:** Intercorrelations between the latent factors and with age as a covariate (model 3).

Factors	Boys				Girls			
	(1)	(2)	(3)	(4)	(1)	(2)	(3)	(4)
Self-movement	1				1			
Object movement	0.96 **	1			0.82 **	1		
Social integration	0.39 *	0.47 **	1		0.00	0.12	1	
Relationship skills	0.16 *	0.20 *	0.20 **	1	0.07	0.16	0.17 *	1
Age	0.44 *	0.71 **	0.12	0.13 *	0.43 **	0.64 **	0.15 **	0.15 *

Note: (1) Self-movement, (2) Object movement, (3) Social integration, (4) Relationship skills; * *p* ≤ 0.05 and ** *p* ≤ 0.01.

## Data Availability

The data presented in this study are available on request from the corresponding author. The data are not publicly available due to ethical guidelines of the Cantonal School Authorities.

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
