# Peer review of "Connection between Social Relationships and Basic Motor Competencies in Early Childhood"

_children, 2021, doi:10.3390/children8010053_

Round 1
Reviewer 1 Report
This is an extreme polished submission. It is difficult to find any faults of note in the presentation, the research methodology and or the conclusions drawn from the findings.
Two comment that I think should be addressed though.
1. Did the sampling strategy have any element of randomisation, or were all preschoolers who were enrolled accepted? It feels like from the reported sample that approximately 10 preschoolers per class were admitted.
2. Was there any evidence for heterogeneity or variance in the data due to the district or preschool? I ask as one of the critical variables (social integration) involves teacher assessment. There certainly appeared to be a greater deal of heterogeneity in the MOBAK rather than the PIQ/KOMPIK. Would be useful to see the range of scores. From the data presented it appears as though the preponderance of responses are in the top 25% of the scale (perhaps explaining the weaker correlations from those variables). Is there perhaps some form of attenuation/ceiling occurring here? I guess the critical question is one of trusting the judgement of the teacher/parent, and how the team validated this component of the dataset.
Aside from that very interesting paper, that is really polished. Good way to end 2020/start 2021.
Reviewer 2 Report
the paper addresses the interrelated nature of development (holistic learning) that is a well established understanding in ECCE. Similar findings are associated with social competence and Executive Function - see Caporaso, J. S., Boseovski, J. J. and Marcovitch, S. (2019) “The Individual Contributions of Three Executive Function Components to Preschool Social Competence,” Infant and Child Development, 28(4). doi: 10.1002/icd.2132. for example.
I am not necessarily best qualified to judge the quality of quantitative research, but I am concerned about the infilling of data using algorithms. I am not convinced this is necessary, given the number of participants and believe that any incomplete data should be excluded from the data set.
I would appreciate a comment in the discussion on the relative strength of the relationship between different factors- this would support those who are less familiar with correlational analysis to understand the impact of the findings.
The context of the paper needs to be clearer from the beginning- i.e. more clearly situated within the Swiss context and some details about the Swill lehrplan for early childhood - particularly with reference to social and physical content of the lehrplan. This would be of important to international readers who work with very different ages/plans.
Some further justification as to why children were excluded from assessing their peer relationships is needed.
Round 2
Reviewer 2 Report
The concerns I expressed in the previous review have been addressed very well, thank you. I do not have any concerns about publication